# Colostrum provision and care of calves among smallholder farmers in the Kaziranga region of Assam, India

Andy Hopker[1]*, Naveen Pandey[2], Jadumoni Goswami[2], Sophie Hopker[1],
Rupam Saikia[2], Amy Jennings[1], Dibyajyoti Saikia[2], Neil Sargison[1], Rebecca Marsland[3]

**1** Royal (Dick) School of Veterinary Studies, University of Edinburgh, Easter Bush Veterinary Centre, Roslin, Midlothian, Scotland, United Kingdom, **2** The Corbett Foundation, Kaziranga Office, Village Bochagaon, Kaziranga, District Golaghat, Assam, India, **3** School of Social and Political Sciences, University of Edinburgh, George Square, Edinburgh, Scotland

* ahopker@exseed.ed.ac.uk

**Data Availability Statement:** All relevant data are within the manuscript and its Supporting Information files.

## Abstract

Smallholder cattle farming in Assamese villages is sub-optimal in terms of calf survivability, growth, age at first service, and milk yield. Proper understanding of the local situation is essential to formulate appropriate, locally driven, livestock keeper education to sustainably improve animal health, welfare and productivity. In-depth interviewing and direct observation were used to understand the farming strategies, husbandry practices and challenges to health and productivity in a cluster of typical villages in the Kaziranga region of Assam, India, where resource use is balanced between the needs of humans and livestock, with competition from wild species. Knowledge of the importance of colostrum consumption by calves is poor. Timely consumption of sufficient colostrum (locally called "phehu") by calves was clearly sub-optimal in the majority of households. The reasons behind this are nuanced, but the practice of collecting colostrum from newly calved cows to make confectionery for human consumption is an important contributory factor. Care of the umbilicus of the newborn is not routine practice in the locality. Local women are the key group assisting with young and sick animals, including cases of simple dystocia and retained foetal membranes. Cows are usually milked once daily, to attempt to balance the needs for milk of household with those of the calf, which can result in suboptimal nutrition for calves. There are clear opportunities to improve animal health and productivity through locally provided farmer education, particularly with reference to colostrum provision, and the engagement of women farmers in any such programme is key to success.

## Introduction

Cows and calves are essential to the livelihoods and culture of smallholder farmers in rural India. Cows are milked for home consumption and sale of milk, providing an essential protein source, particularly for children, and allowing a small but regular cash income. Calves can be used to expand herd size or can be raised for sale, which provides money to cover family

**Funding:** This work was funded as part of the Royal (Dick) School of Veterinary Studies Indian Veterinary Education Project. The funders had no role in study design, data collection and analysis, decision to publish, or preparation of the manuscript.

expenses, or to pay for farm improvements. Castrated male calves of sufficient quality can be trained as draft oxen to provide agricultural power, which increases their value if sold. Cattle dung is an important fertiliser for crops, and in some areas is dried to use as fuel. Cows have important social and cultural roles, with their own place within the household. As well as having great economic importance, cows are linked to social status, and have religious and spiritual significance, being respected, venerated and protected in Hinduism. Owners usually feel great affection for their animals, which are often considered part of the family, and owners feel strong responsibilities towards them [1].

In order to receive the best start in life, and have the best chances of surviving and then going on to lead a healthy, productive life: a calf needs to receive maternal colostrum; be protected from the elements, predators, and from disease; and receive adequate nutrition. The importance of early life in neonatal calves is known to have a strong influence on future growth, health and productivity [2]. Consumption of maternal colostrum by the calf as soon as possible after birth has long been recognised as essential for passive transfer of immunity from the dam to offspring through gut absorption of IgG contained within the colostrum [3]. Failure to consume sufficient colostrum in the hours immediately after birth results in failure of passive antibody transfer (FPT) and is associated with poor health and survival outcomes [4,5,6,7].

It is recommended that a calf should ingest 10% of its body weight as colostrum within the first 6 hours of life, an illustration of how this can be calculated is as follows: A calf should achieve serum total protein of >60g/L by 48 hours old [8], which is a serum IgG of 25g/L [9]. The mean IgG concentration of colostrum from Holstein cows is 68.8 g/L [10], and absorption of IgG from colostrum by the calf's gut is approximately 22% efficient [11] therefore each litre of colostrum fed contributes approximately 15g of serum IgG to the calf.

A typical Holstein weighing 45kg has an approximate plasma volume of 4 litres (9% of bodyweight) and thus requires 100g of IgG which will be delivered by approximately 6.5 litres of colostrum (15% of bodyweight) in the first 24 hours, given as 4.5 litres (10% of its body weight) within the first 6 hours of life and a further 2.25 litres (5% of body weight) in the next 18 hours. It is considered that colostrum ingestion should start within the first hour after birth, and if administering by stomach tube, generally no more than 2 litres should be given at a single feed to a European breed calf {5,7}. It should be noted that the above illustration is an example, and there is variation in the IgG content of maternal colostrum, dependant on a number of factors, including dry cow nutrition, parity, season, colostrum yield, and vaccination status of the dam [12, 13]. As significant variation can occur in adequacy of passive transfer both within and between farms, the ideal system is to assess both colostrum quality and calf serum IgG on an individual animal basis. This can be readily achieved using a standard Brix refractometer [14, 15]. Access to veterinary professionals and paraprofessionals, equipment, and the pre-existing attitudes of livestock keepers to medical procedures can complicate this goal, in some regions of rural India there is resistance to allowing the blood sampling of apparently healthy bovines.

Despite the wealth of published data on the subject, FPT remains a major problem in both intensive and extensive Western farming systems, with rates of FPT in US dairy farms estimated as between 20% and 40%, depending on the definition of FPT, and mortality of dairy calves linked to FPT estimated as 8–25% [16]. While problems affecting Western farming systems and breeds animal cannot be directly related to the situation in farming in India, it is likely that many similar challenges exist.

The situation regarding colostrum ingestion and eventual outcomes is less well documented on the Indian Sub-Continent than in intensive Western farming systems. A study of small, medium and large farms in Uttar Pradesh found only 12% of dairy buffalo farmers reported

feeding colostrum within 2–3 hours of birth [17] and a study of small holders in Bihar found only 15% of rural and 10% of peri-urban cattle keepers reported feeding the calf colostrum prior to the passage of the placenta by the cow, as they believed colostrum feeding delayed expulsion of the foetal membranes by the cow [18]. In the Rahim Yar Khan District of Pakistan a study of small, medium, and large dairy farmers found only 12% of farmers reported feeding colostrum to buffalo calves prior to the passage of the placenta [19]. Each of these three studies also reported a belief among farmers that early colostrum feeding caused diarrhoea to occur in calves. It should be noted that, in some areas of India, traditional attitudes advocate colostrum rejection prior to initiating breastfeeding of human babies, in the belief that the thick yellow colostrum is stale, unpalatable or unclean [20]. Interviews with Gir cattle keepers in villages of the Ajmer district of Rajasthan were slightly more encouraging, with 51% of participants reporting allowing the calf to suckle colostrum within two hours of birth, 12% within 2–4 hours, and 36% after the dropping of the placenta [21].

A number of factors influence how soon after birth a calf ingests colostrum. These include ease or trauma of birth, which is influenced by management of dystocia in a prompt, competent and welfare friendly manner; access of the calf to its dam; human assistance or interference with nursing; disturbance by people or animals; maternal health influencing supply of colostrum and the ability to nurse the neonate; and provision of thermal protection for calves in cold conditions [7, 22]. Ensuring adequate nutrition of the dam in the pre- partum period is essential for the production of good quality colostrum [23].

Milking of colostrum for human consumption will further reduce the available supply for the neonate. This activity is widely practiced in many areas of India. A study in the Kanha region of Madhya Pradesh found that the majority of participants at 37 out of 38 livestock keeper meetings in different villages reported taking quantities of colostrum (referred to as "Khees" or "Cheek" in that region) for home consumption from newly calved cows and buffalo [24]. Typically, the colostrum is boiled with sugar to make confectionery called "ginna" in Hindi, with various other names in other Indian languages [25]. On the Indian Sub-Continent some people ascribe benefits to human health to the consumption of bovine colostrum [26].

Other interventions during the perinatal period, such as reducing the challenge of infection through hygiene and care for the open navel (umbilicus); protection from extremes of weather; and allowing the cow and calf to bond, will also affect calf survival, growth and productivity outcomes. Beyond the perinatal period the care and nutrition of the calf in early life will affect its health, growth and future productivity [27], and of particular importance is the available supply of milk and fodder to the calf.

Sustainable improvements to calf care have the potential to improve rural prosperity by increasing the survivability and future productivity of these animals. However, to realise such benefits it is essential to fully understand the current situation, and the opportunities for locally driven development programmes. Here we report the investigation of routine care new born and young calves in a typical Assamese village, exploring the social, cultural, economic and environmental drivers for smallholder livestock keeper behaviours by rich data collection through in -depth interviewing and observation of village life.

## Materials and methods

A series of interviews was carried out over a two-week period in a cluster of three villages bordering the Kaziranga National Park, in Assam, North East India. The data collection process was designed and reported in accordance with the COREQ guidelines [28]. In depth interviews were carried out at the homes of 17 smallholder farmers. Farmers were either interviewed alone, or together with one or two family members or friends. Additional family and

friends frequently came and went during interviews, sometimes joining in for a time. The interviews typically lasted two to three hours and employed open questions, with follow up questions as indicated, to create as informal and conversational an interaction as was realistic in the circumstances. The interviews covered farming practices, with particular reference to ruminant livestock keeping, and perceived challenges to health and productivity. Attitudes to farming, and the underlying socio- economic and environmental factors were also explored, including discussions about the role of education, alternative livelihood strategies, the effect of the proximity of the National Park, and the future of farming in the village. Interviews were conducted in Assamese by a local Assamese community worker (JG) who was already known to the participants, and a UK veterinary surgeon (AH), an experienced clinician and researcher. Participants were encouraged to freely discuss any and all aspects which were of importance or interest to them. All participants were volunteers, introduced to the project by a friend or relative, and care was taken to include animal keepers from a variety of social levels, families, and income levels within the village, and to ensure a balance of gender and age. No household declined to take part in the study. The aims and scope of the study were explained to the participants, and informed verbal consent recorded. The use of signed consent forms was not appropriate due to local mistrust of signed documents and varying levels of literacy. Hand written notes were made during interviews. Use of voice recording was not acceptable to the participants. Following the completion of the interview schedule the participants attended a community meeting to discuss the findings of the study and possible sustainable mitigating strategies. Interview notes were transcribed and NVIVO 11 (QSR International Ltd) was used to build a node structure, to which responses were coded for analysis. The manuscript was prepared in accordance with the Synthesis of Guidelines for Reporting Qualitative Research [29]. Only the information relating to care of calves, and the relevant background to this, is presented in this manuscript.

## Ethical approval

This work was carried out under ethical oversight and approval from the Royal (Dick) School of Veterinary Studies, University of Edinburgh, Human Ethical Review Committee (approval number HERC 47 17). All participants were volunteers, no participant declined to take part in the study.

The aims and scope of the study were explained to the participants, and informed verbal consent, including for the use of photographic images, was recorded on the interview paperwork, witnessed by both members of the interview team. The use of signed consent forms was not appropriate due to local mistrust of signed documents, and varying levels of literacy in the area, and this use of verbal consent was approved by the Royal (Dick) School of Veterinary Studies, University of Edinburgh, Human Ethical Review Committee.

## Study area

The area of study is a corridor of agricultural land and villages which varies in width between a few hundred metres and a few kilometres bordered by the Kaziranga National Park to the north and the National Highway 37 to the south. The National Park spans the districts of Golghat and Nagaon and incorporates the Brahmaputra River to the north. Kaziranga National Park, was officially recognised in 1974, having been designated a forest reserve since 1905. It is a UNESCO World Heritage Site. Kaziranga is home to two thirds of the world population of the Indian one horned rhinoceros, as well as wild Asian elephants, water buffalo, tigers and deer. These animals are able to leave the park area and enter farmland and villages. The area is

subject to a monsoon climate with a seasonal inundation, typically lasting between 10 and 20 days, usually occurring in the months of June and July.

The study villages are directly adjacent to the park boundary, which adjoins it on two sides at a distance of less than 300 metres. There are no settlements or agricultural activity within the park itself, and entry into the park is prohibited, other than through a strictly controlled safari permit system. However local people undertake agriculture activities right up to the park boundary, and wild species frequently leave the park and penetrate agricultural land and villages. The villages are typical for those in the region in terms of agricultural practices, education levels and cultural practices.

## Results

In–depth interviews were carried out between 16th and 23rd March 2017. 17 interviews encompassed 18 households. Interviews were carried out at the participants' homes. Eight interviews were with a single main respondent: five men and three women; nine interviews were with a pair of main respondents: four married couples, two adult daughters with their mothers, two adult sons with their mothers, and one pair of brothers. One of these interviews was with three respondents, a married couple and a neighbour representing a separate household. Additional family members or friends frequently joined interviews for a time. The group interviews, including the contributions of visitors, yielded very interesting conversations and perspectives. The stated age range of the main participants was 25–65 years.

### Agriculture, livestock, and farmers

Almost all village inhabitants identify themselves as farmers, including those who have others jobs, including teachers; shop keepers; workers in the tourist and other service industries, including middle managers; and trades people. Farming is usually a mixed enterprise, concentrating on cultivation, with livestock using marginal resources, some of which would otherwise be wasted. Livestock provide protein for household diets, and cash income which is more flexible than the income from cultivation which is concentrated at harvest times. In order to properly understand the role of livestock in the village and their importance, it is essential to take a holistic overview of the village, its inhabitants, and their farming practices and economic strategies.

Typically, people are smallholder farmers cultivating rice paddy as their primary crop and mustard as the main secondary crop, alongside vegetables grown for home consumption and sale. A single seasonal crop of rice plus a crop of mustard grown on the same land following the rice harvest is possible annually using natural irrigation. For those farmers able to afford to purchase and fuel an irrigation pump, an unseasonal rice crop is also possible. However, due to patterns of crop raiding by wild animals, which is a significant cause of crop losses in the area (participants report losing between 20% and 50% of their annual rice harvest to crop raiding by wild species), some farmers are growing the unseasonal crop only in an attempt to exploit patterns of crop raiding. Vegetables and bananas are generally grown in gardens adjacent to dwellings. Cattle, goats, chickens, ducks and pigeons are kept, variously for milk, draft power, sale, eggs, and meat.

Cows are typically native breed animals of small stature, with adult cows weighing approximately 200 – 250kg. A small number of animals are crossbreeds, typically of Jersey cattle. Age at first calving is typically between three and four years old. Nutrition of cattle is primarily pastoral grazing, with dry paddy fields, uncultivated land, field and road edges being used (Fig 1). Adult animals are frequently tethered, while youngsters usually roam freely around their mothers. At certain times during the agricultural cycle, adult animals graze untethered on

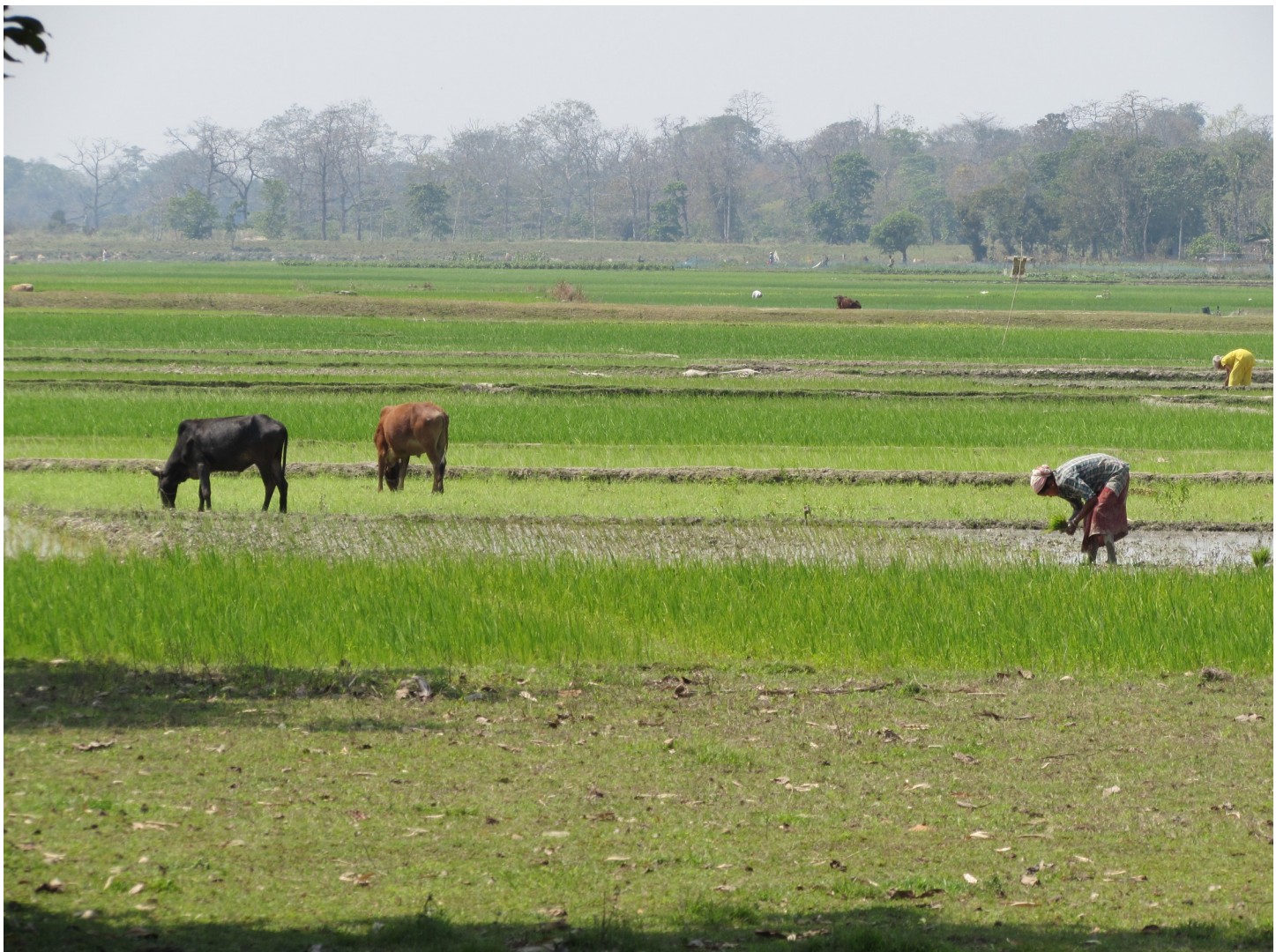

**Fig 1. The close association of people, their animals, cultivated crops, and homes; and the interface with the wild environment.** Livestock graze dry paddies while others are in cultivation. Draft power and mechanical cultivators are both employed, but there is still a high level of manual labour. Here young rice plants, which have been grown in nursery paddies are being planted out by hand. The village is largely located within the tress and the boundary of the Kaziranga National Park lies less than 200 metres beyond the visible treeline.

open areas and uncultivated paddies. Conserved forage is mostly rice straw, and banana plants are felled and chopped for fodder as part of the replanting cycle, especially during the wet season. Most cattle keepers describe giving their animals "cooked food", usually a mixture of rice husk, broken rice, vegetables, water and salt; cooked down and served warm. Water from hand pumps or wells is usually presented to cattle morning and evening, and buckets of water are carried out to animals at tethered grazing. Some owners take animals to the river or areas of standing water to drink, free grazing animals can also access water in this way. Animals are housed during hours of darkness (Fig 2) primarily for protection from predators, mostly tigers in this area. Agricultural power is provided by both draught oxen, and machinery such as tractors and cultivators, both of which are usually rented with an operator on a per job basis.

All households in the village undertake agricultural activities, while many also pursue additional income streams such as weaving; daily labour; trade, such as carpentry; small shops; and

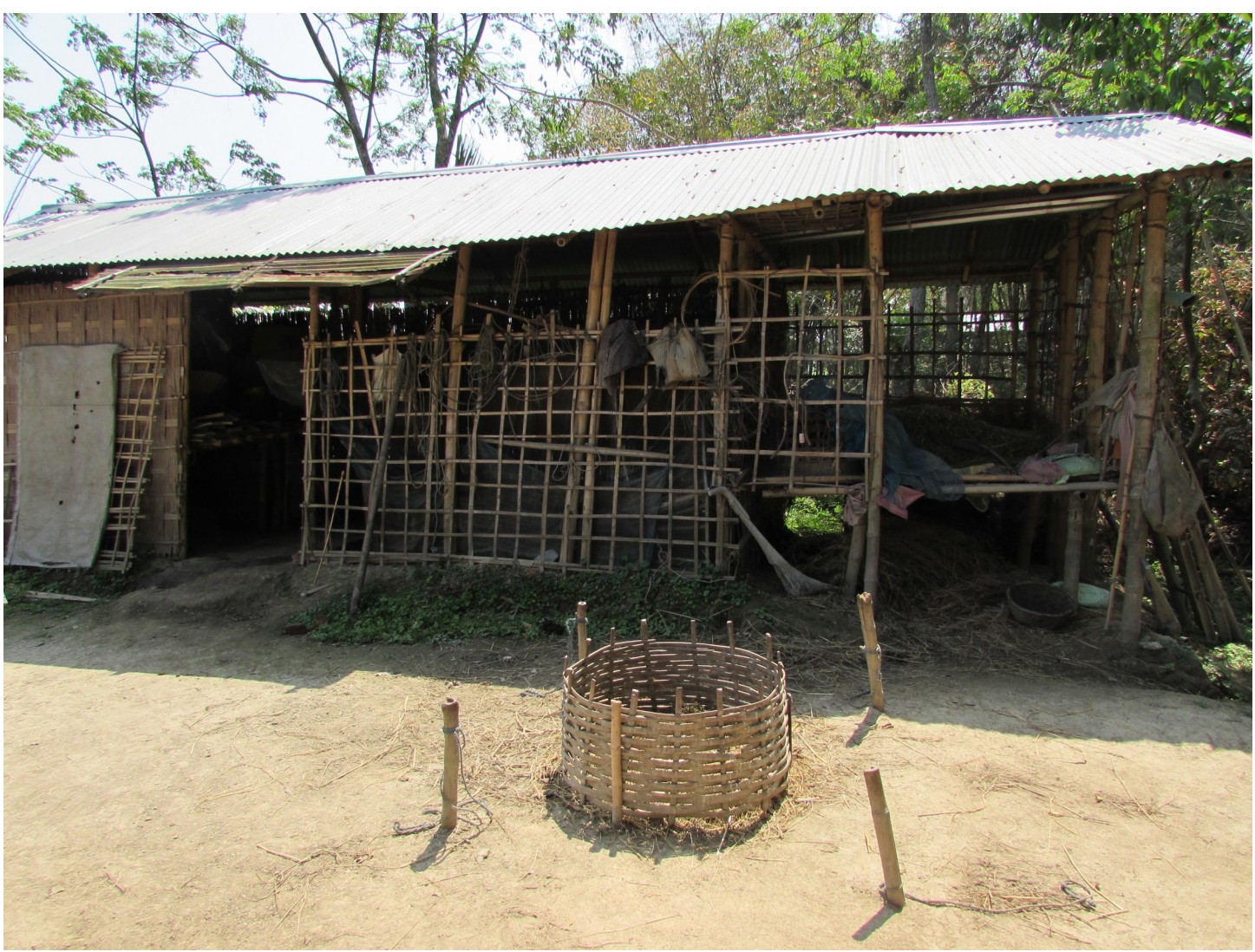

**Fig 2. Typical livestock accommodation.** Built from bamboo with a corrugated steel roof for protection from the rain, this structure is not entirely secure against large predators such as tigers. The central area of the accommodation might be used for cattle and goats, or goats could be accommodated elsewhere in a raised structure. The mud floor of the accommodation is slightly raised for drainage, important if hygienic conditions are to be maintained. The right section is a store for conserved fodder, mostly rice straw, the raised left section is a store for firewood. Poultry will also share this accommodation, and some farming implements and fishing tools stored inside. Note the ring feeder for fodder with four tether points in front of the structure.

small trading businesses. All children receive primary education at the village school, after which most attend a nearby high school, though not all complete the programme. A very small number of young people are able to undertake studies at a university. Few smallholders have formal education in agriculture. As a result of these complex multiple income strategies and the challenges of living in this landscape, and the need to divide nutritional resources between people and animals, there is limited time and means available for the care of calves, despite their important contribution to the household.

The eighteen households participating in this study own a total of 35 cows between them. In all cases except household six these cows were non- descript native crossbreed cattle. Household 6 owned Jersey cross- breed animals. In 2016–2017 those 35 cows gave birth to 31 calves, of which 6 died before one year old. The causes of death of these calves were described

respectively by the keepers as: weakened and died around 15 days old; calf stopped eating and then died; black quarter; died during the flood; eaten by a tiger; died suddenly around 6 months old.

**Routine care of newborn calves.** All participants were genuinely and earnestly concerned for the well-being of both their cows and calves, spoke fondly of their animals and were indicated a strong sense of responsibility towards them. Participants were keen to explain the steps that they took to ensure that their calves received the best start in life. These husbandry techniques are summarised below in Table 1.

**Table 1. Care of newborn calves.**

| Partici-pant | Colostrum taken for house? | How much taken? (litres) | Colostrum routine | Clean calf? | Navel care | Rice straw in mouth? | Comments by farmer (respondent's gender M/F, age, and relationship) |
|---|---|---|---|---|---|---|---|
| 1 | Yes | 0.5–1 | After placenta falls (1–2 hours), milk off the colostrum, then the calf drinks | | | yes | Cow not milked again until calf 7 days old. (M, 45) |
| 2 | Yes | 0.5–1 | After milking the colostrum, then put the calf on the teat | yes | Wash with mustard oil for 5–6 days and remove flies. Himax™ applied if flies come. | | Next milking 15 days (M, 49 + F, 44? Husband and wife) |
| 3 | Yes | 1 | Take 70% of the colostrum, then allow the calf to drink | yes | | yes | Check movement of tongue. Cut tips off hooves. Next milking 7 days, cow and calf together at all times until then. (F, 30 + F 55? Daughter and mother) |
| 4 | Yes | 0.5–1 | Yellow colour faded in milk before the calf drinks. | yes | | yes | Yellow colour faded in milk before calf drinks. Start fire to warm calf if born in winter. Next milking 5–6 days, cow and calf together at all times until then. (M, 27) |
| 5 | Yes | 1 | Milk all four teats (for colostrum), leave one third of colostrum for calf | | | | Clean calf. Make fire to warm calf in winter. Help to stand. Next milking 5–6 days, cow and calf together at all times until then. (M, 39 + F 37? Husband and wife) |
| 6 | Yes | 0.5 | Put the calf to suck, it drinks until full, then milk out the remainder. | yes | | | Put a tiny cup of mustard oil in calf's mouth. Milk again the next day. (M, 49 + F, 65? Son and mother) |
| 7 | No | | "Get the calf to drink as soon as possible. . .. Take no colostrum." | | Keep clean to prevent flies | | "If calf stumbles when it starts to walk, steady it and take to teat. Can use fingers to encourage to suck and open mouth if required. "Make a fire to warm calf and boil water. . . Give cow warm water to drink and placenta falls away quickly." (M, 42 + F, 65. Son and mother) |
| 8 | Yes | 0.5 | Draw colostrum from all four teats. "The calf drinks after milking is finished- approximately 1–1.5 hours old if he stands on own. Assist the calf to stand and suckle if necessary" | | | | Give cow cooked food- broken rice, vegetables, banana tree. Give some colostrum sweets to cow and calf First 7–10 days cow and calf together at all times. (F, 25) |
| 9 | Yes | 1 | Milk all four teats half empty. The calf then drinks the other half. The calf drinks when it is about 30 minutes old. | | | | (M, 43) |

*(Continued)*

**Table 1.** (Continued)

| Partici-pant | Colostrum taken for house? | How much taken? (litres) | Colostrum routine | Clean calf? | Navel care | Rice straw in mouth? | Comments by farmer (respondent's gender M/F, age, and relationship) |
|---|---|---|---|---|---|---|---|
| 10 | Yes | 0.5 | Help the calf to stand and drink. Open mouth and put on teat.<br>Allow the calf to drink for 15–20 minutes, then milk colostrum for the house. We do not milk the udder empty | yes | | | "It can be very difficult to feed new born calves."<br>"If it drinks the first milk it will become strong and healthy. If it does not drink it will become thin and weak. It might die if it does not get the yellow milk." (F, 33) |
| 11 | Yes | 1 | The calf stands after 1.5–2 hours, take to cow and help to feed.<br>Calf drinks first, for 2–3 minutes, then we milk the cow. After 1–1.5 hours we allow the calf to drink again. | Yes | | | "Clean the calf, this is a woman's job, clean face and nose, sometimes blow air in nostrils. Take calf out of shed to a sunny place if born in the day. The calf stands after 1.5–2 hours, then take (the calf) to cow and help to feed. Open its mouth and put to the teat, gently rub its head and then it starts to suck on its own. Sometimes you put your own finger in its mouth to encourage it to suck."<br>"The calf will drink full in 2 minutes."<br>"Someone looks after the calf for the first 12 hours to check that it is drinking"<br>Next milking 10 days. Calf and calf are together all that time. (M, 58) |
| 12 | Yes | 1 | "Milk out first colostrum as soon as cow stands up. Don't milk empty, take about half, so if the cow normally gives 2 litres, take one litre of colostrum. Then we help the calf to suck." | | | | "Look after the calf, if he can't stand or suck, help him. If unable to drink- feed with bottle, 100 – 200ml, if not fed in 2 hours."<br>Give cow food including pepper and boiled vegetables (M, 35) |
| 13 | Yes | 0.5 | When the calf tries to stand- help it to feed from mother.<br>The calf drinks for 15–20 minutes, then milk the cow "Estimate amount of milk from the size of the cow- half for the calf." | yes | | yes | If winter make fire to keep the calf warm.<br>Give the cow hot water to drink, bamboo leaf and sugar cane leaf to eat- then the placenta falls quickly.<br>Don't milk cow for 7 days, cow and calf together the whole time. (M, 52 + M 55? Brothers) |
| 14 | Yes | 1 | Before the calf drinks milk out 1 litre from the cow using all 4 teats. The calf drinks half after the cow is milked. The cow is milked 1 hour after the calf is born, the calf first drinks at 1.5 hours old. | yes | Clean navel with coconut oil and protect from birds. | | "Not drinking first yellow milk prevents calf from having diarrhoea."<br>Feed the cow- cabbage or any available vegetables and rice husk.<br>"Observe carefully and check that the calf is drinking."<br>"Make also a lucky string of colostrum beads (from colostrum sweets) and put round the cow and calf's necks." (F, 34) |
| 15 | Yes | 0.5–1 | "Help the calf to drink- when he can stand, he drinks.<br>The calf drinks 0.5 litres, then we milk for house." | | | | "Look after the calf, help it to drink. Make a fire if it is winter."<br>Give cow cooked food- papaya, vegetables, concentrate powder mix (mineral supplement).<br>Next milking 10–12 days, the cow and calf are together all the time. (F, 45 + M 45? Wife and husband) |

(Continued)

 

**Table 1.** (Continued)

| Partici-pant | Colostrum taken for house? | How much taken? (litres) | Colostrum routine | Clean calf? | Navel care | Rice straw in mouth? | Comments by farmer (respondent's gender M/F, age, and relationship) |
|---|---|---|---|---|---|---|---|
| 16 | Yes | <1 | Help the calf to stand, help it to drink from mother- usually the calf drinks after half an hour. The calf drinks until full, then we take colostrum | Yes | | | (M, 44 + F, 38. Husband and wife) |
| 17 | Yes | <1 | The calf drinks until full, then we take colostrum | yes | | | (M, 55) |
| 18 | Yes | 1 | The calf drinks first, for 10–30 minutes. Usually starts 20–30 minutes after birth | yes | Check navel, if maggots found, put himax, or tobacco, might cover with a bandage. | yes | Feed the cow banana leaves and warm water, especially if placenta did not pass. (F, 25 + F, 55? Daughter and mother) |

Table notes: Gender, ages and relationship of main interviewees noted; ? indicates age estimated either by participant or researchers.

## Provision of colostrum

Almost all (17 out of 18) participants described milking colostrum, locally called "phehu", from the new calved cow for home consumption, boiling it down with sugar to make sweets, which are also called "phehu". Eight participants said that they let the calf drink first, however, one of these stated that 'the calf drinks full in 2 minutes', and that 'milking could begin after that'. Nine participants said that they milked off colostrum before the calf drank. Only one participant stated that they take no colostrum for home consumption. While all participants stated that they wanted to give their calves the best start in life, there was a variable understanding of the role of colostrum, as demonstrated by the quotes below:

"*If it (the calf) drinks the first milk it will become strong and healthy. If it does not drink it will become thin and weak. It might die if it does not get the yellow milk.*"

(Participant 10, female, 33 years old)

"*Get calf to drink as soon as possible. . .. Take no colostrum.*"

(Participant 7, male, 45 years old)

While other participants understand the situation differently:

"*Not drinking first yellow milk prevents calf from having diarrhoea*"

(Participant 14, female, 34 years old)

"*yellow colour (of colostrum should be) faded in milk before calf drinks*"

(Participant 4, male, 27 years old)

Participants described local cows as giving 1–2 litres of colostrum on the first day, typically 1.5 litre, although this could be an underestimate as the calf might be able to extract more through suckling than animal keepers can by milking the cow. Participants describe taking between 0.5 and 1 litre of this for home consumption, leaving 0.5–1 litre for the calf. Calves

born to native cattle in the region typically weigh 15–20 kg at birth. Based on the calculations described in the introduction might be expected to require between 1.5–3 litres of colostrum in the first 24 hours of life. As previously discussed, many factors could influence the true colostrum requirements of these calves, however it is likely that the majority of calves born in this locality are receiving insufficient colostrum

**Drivers behind making colostrum confectionery.**    Participants stated that it is their custom to share any food that they have in abundance with their neighbours, whether that is vegetables from their gardens, a dish which has been prepared, or sweets that have been made. Participants asserted that sharing in this way is important to them as a community, reinforcing community bonds, and is linked to sharing responsibilities and labour, and helping each other. The birth of a calf is a cause for celebration, and sweets are made from the cow's colostrum because colostrum is available, and because the participants like the sweets, rather than because of a special cultural or religious importance assigned to the sweets greater than the sharing of other foods. Some participants describe their attitude to phehu confectionary:

"*This is a thing that is freely given, it is not expected. It is our custom to share with our neighbours, tomatoes, sweets, some dish (of curry), it is the same thing. . . The sweets are not special in themselves. . . If a family's cow gives birth, I do not expect them to bring me sweets.*"

(Participant 3, female, 30 years old)

However, the importance of the equal treatment of immediate community members was emphasised:

"*If no sweets were given, I would not be offended, but if sweets were given to one neighbour and not to another, then that fellow who did not receive the sweets, he would be offended*"

(Participant 17, male, 55 years old)

In addition to these statements, one participant did refer directly to "a string of lucky phehu beads" and another described a specific pattern of milking the cow a little and pouring it into the river between days 3 and 11 post-partum, suggesting a more nuanced relationship with bovine colostrum. Participants did not mention any special benefits to human health resulting from the consumption of bovine colostrum.

**Physical assistance of the calf.**    Participants described helping their calves to stand and suckle. The time from birth to first suckling varied, with the majority of respondents stating that the calf should drink as soon as it stands, typically estimating this time to be 1–2 hours postpartum. Three participants stated that the calf normally drank 30 minutes after birth, the remaining seven participants did not offer a time postpartum at which they considered it normal for suckling to start. Eight participants specifically stated, unprompted, that they help the calf to stand and suckle as part of their normal routine. One participant stated that they bottle feed a calf that has not suckled, giving 150 – 200ml. Five participants stated that they tie a band of rice straw around the calf's head and through its mouth to stimulate suckling behaviour.

"*Tie rice straw in the mouth and round back of the neck or behind ears, then check suck. Leave in place until breaks or mother removes. . . the straw breaks itself after approximately 30 minutes.*"

(Participant 1, male, 45 years old)

Table 2. Milking practices.

| Partici-pant | Colostrum taken for household consumption? | Days post-partum to start milking? ie days calf full drinking | Once or twice daily milking? | How many teats milked? | Milk empty? | How much milk / cow/ day | Whose job usually? | Comments by farmer (respondent gender M/F) |
|---|---|---|---|---|---|---|---|---|
| 1 | Yes, 0.5 -1L | 7 | Once | 4 | Yes | | | |
| 2 | Yes, 0.5 -1L | 15 | Once | 4 | No | | | |
| 3 | Yes, 1L | 7 | Twice after 3 months pp | 2 | Yes | | All | |
| 4 | Yes, 0.5 -1L | 5–6 | Twice after 15 days pp | | | | All | The calf is able to drink for 30 mins in morning before milking (M) |
| 5 | Yes, 1L | 5–6 | Once | 4 | | 1–1.5 L | All except husband | My husband is a teacher so he has no time to help with the cow and calf in the morning. (F) |
| 6 | Yes, after the calf drinks full | 2 | Twice | 4 | Yes | | | Mastitis occurs if the cow is not milked empty. (M) (Note: this household owns Jersey cross breed cows) |
| 7 | No. The calf full drinking only | 3–7 | Twice after 15–30 days pp | 4 | no, 75% | | | |
| 8 | Yes, 0.5L | 7–10 | Once | 4 | no | 1 L | Husband | |
| 9 | Yes, 1L | 10–12 | Once | | | | Wife | |
| 10 | Yes, 0.5L | 30 | Once | 4 | no | 1.5 L | | "If (the calf) drinks the first milk it will become strong and healthy. If it does not drink it will become thin and weak. It might die if it does not get the yellow milk." (F) |
| 11 | Yes, 1 L | 10 | Once | | | 1–1.5 L | Wife | "My wife is in charge of the calf and milking things." (M) |
| 12 | Yes, 1L | 3(11) | Twice after calf starts grazing (15–30 days) | 4 | no | 1.5 L | All | The cow is milked a little every two days from day 3 to 11 and the milk is poured in the river (for religious reasons)... Only from day 12 is milk taken for consumption or sale. (M) |
| 13 | Yes, 0.5L | 7 | | 4 | no, 50% | | | The alf must drink as much as possible, otherwise it becomes weak.(M) |
| 14 | Yes, 1 L | 10 | Twice after 30 days | 4 | no | 1–1.5 L | | "The cow is grazed but not tied when she has young calf." (F) |
| 15 | Yes, 1L | | Once | 4 | | 1–2 L | Lady | |
| 16 | Yes, <1L | 10 | Once | 4 | | 1–1.5L | Wife | |
| 17 | Yes, <1L | 12 | Once | 4 | Yes, after first 30 days | | Husband and wife | |
| 18 | Yes, 1L | 10–11 | Once | | | 1 L | | |

**Table notes:** Not all respondents answered/ were able to answer all questions; pp = postpartum

**Cleaning and navel care.** Eleven participants stated that they clean the calf after birth, and four that they specifically clean the navel and take steps to protect it, primarily from fly strike by topical applications or bandaging. No participant discussed the navel as a potential portal for the ingress of infection, but there was concern that the unhealed umbilicus might attract flies, leading to fly strike (myiasis).

**Protection from the elements.** Five participants described making a fire to warm the newborn calf if the weather is cold. One participant recommended placing the calf to lie in the sunshine to warm it following birth.

**Milk and milking.**    Following the initial peri-natal period, when deciding how much milk to take from their cows, smallholders face a delicate balancing act between the needs of their calves for growth and health, and the needs of their family for nutrition and daily cash income from milk sales (Table 2).

The cow and calf are taken to graze soon after birth (Fig 3). Most participants described a period of time postpartum for which the cow and calf were together at all times; the calf able to drink freely, the cow not milked, and possibly be untied, to enable her to tend to and protect her calf. Only one participant reported that they start to milk their cow on the first day post-partum. One participant reported starting to milk their cow on the second day postpartum; one between 3–7 days postpartum; two at 5–6 days; three at 7 days; seven after 10–12 days; one at 15 days, and one at 30 days post-partum. One participant did not state when they start to milk their cow.

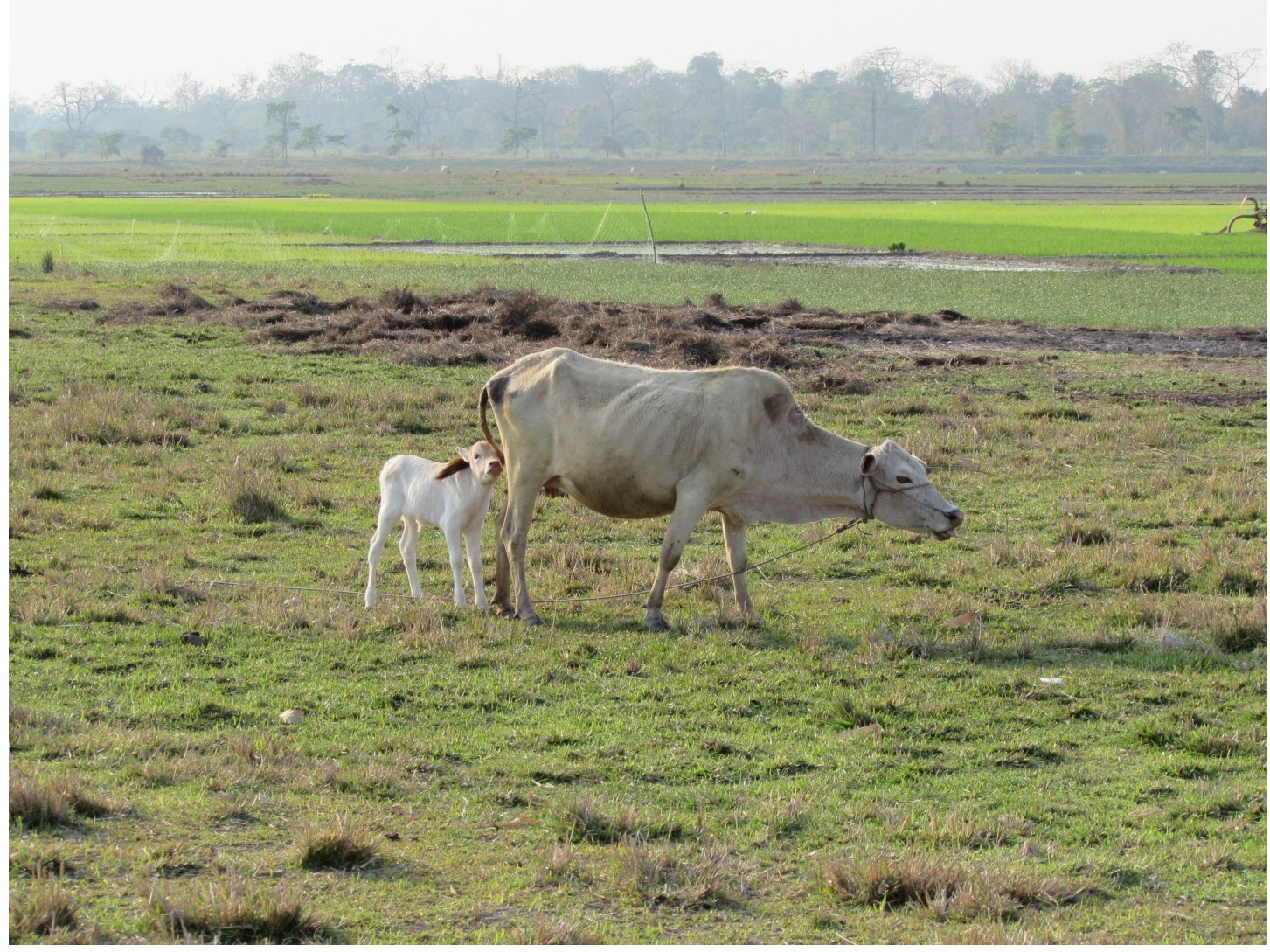

**Fig 3. Cows and calves taken to the fields to graze soon after birth.** It is the local practice in this area that the cow and calf remain together, rather than the calf remaining in the dwelling in the day while the cow is grazed, as is done in some areas of the Indian sub-continent. This co-grazing management allows greater opportunity for the calf to suckle, which is better for calf growth, however it reduces the potential to milk the cow a second time in the evening, affecting household milk yield. While the calves in the field might be at increased risk of infestation from parasites present in the grass, mud or water's edge, they are at much lower risk of common hygiene related infections which result from the build-up of pathogens within livestock accommodation. This cow is tethered to prevent her from straying into the planted paddies. Note the diesel-powered irrigation pump on the right side of the image, necessary for growing this unseasonal rice crop, an example of wealth inequality in this society.

Most commonly cows were milked only once daily, usually in the morning. Eleven participants reported milking once daily throughout the cow's lactation; four reported milking initially once daily, then twice daily after 15–30 days and one twice daily after 3 months. Only one participant reported milking twice daily from day 2, this household was the only one in the study keeping Jersey cross-breed cows. One participant described that from day 3 until day 11, his father would draw a little milk from the cow every second day and pour it into the river or place it under the ground or beneath cow dung. The cow was then milked from day 12 onwards. The participant stated that his father believed that the belief that the cow would subsequently milk better, but the participant seemed less sure of this himself.

"*After milking the phehu the cow is next milked on the third day, and the milk is thrown in the river. Again on the 5ᵗʰ day the cow is milked again and the milk thrown in the river. This is done every two days until 11 days. Sometimes milk is placed under the earth or cow dung. People avoid stepping on the milk. My father does this, it is religion.*"

(Participant 12, male, 35 years old)

Four participants stated that they milked the udder completely empty; six that they left some milk in the udder for the calf. Those participants that milked twice daily often discussed starting twice daily milking once the calf started grazing. Responsibility for routine milking and calf care was more commonly taken by women than men, but many people discussed sharing this work between all family members.

A participant described the typical milking routine for native cattle (Fig 4):

"*The calf is separated from the cow overnight. In the morning send calf to cow and allow it to drink for a few minutes, this starts let down (of milk). Then remove the calf and milk cow empty. Return calf to the cow, there is a second let down, remove the calf and milk again. Then the calf gets the rest of the milk. The cow then goes to the field, and the calf goes with the cow for the rest of the day*"

(Participant 1, male, 45 years old)

The typical pattern is to allow the cow and calf to be together at all times for approximately the first ten days of life. After that the calf is separated from the cow (using a small pen within the cowshed) at some time between midnight and 2 am, the animals having been housed at nightfall, which occurs between 4.30 and 5 pm in the region, for protection from predators. As the calf grows, and begins to ingest more fodder the time of separation is moved earlier to midnight, which then continues in a stepwise fashion until it reaches about 7 pm.

**Care of cows during the peri-partum period.** All participants expressed great affection for their cattle and were equally concerned with the wellbeing of the cow as the calf. There was recognition by respondents that good care of the cow during the perinatal period was beneficial for her ability to raise the calf, but also that it was the right thing to do for the welfare of the cow, which is considered as a family member.

**Dystocia.** Dystocia, if poorly managed, can result in major birth trauma for both cow and calf and can result in reduced and or delayed colostrum consumption, and hinder the formation of the cow-calf bond. Five participants raised the subject of dystocia during the interview. The prevailing opinion was that straightforward cases of dystocia are best dealt with promptly, by farmers themselves, and that women were more knowledgeable and skilled than men in the delivery of baby animals, and in the care of young or sick animals generally.

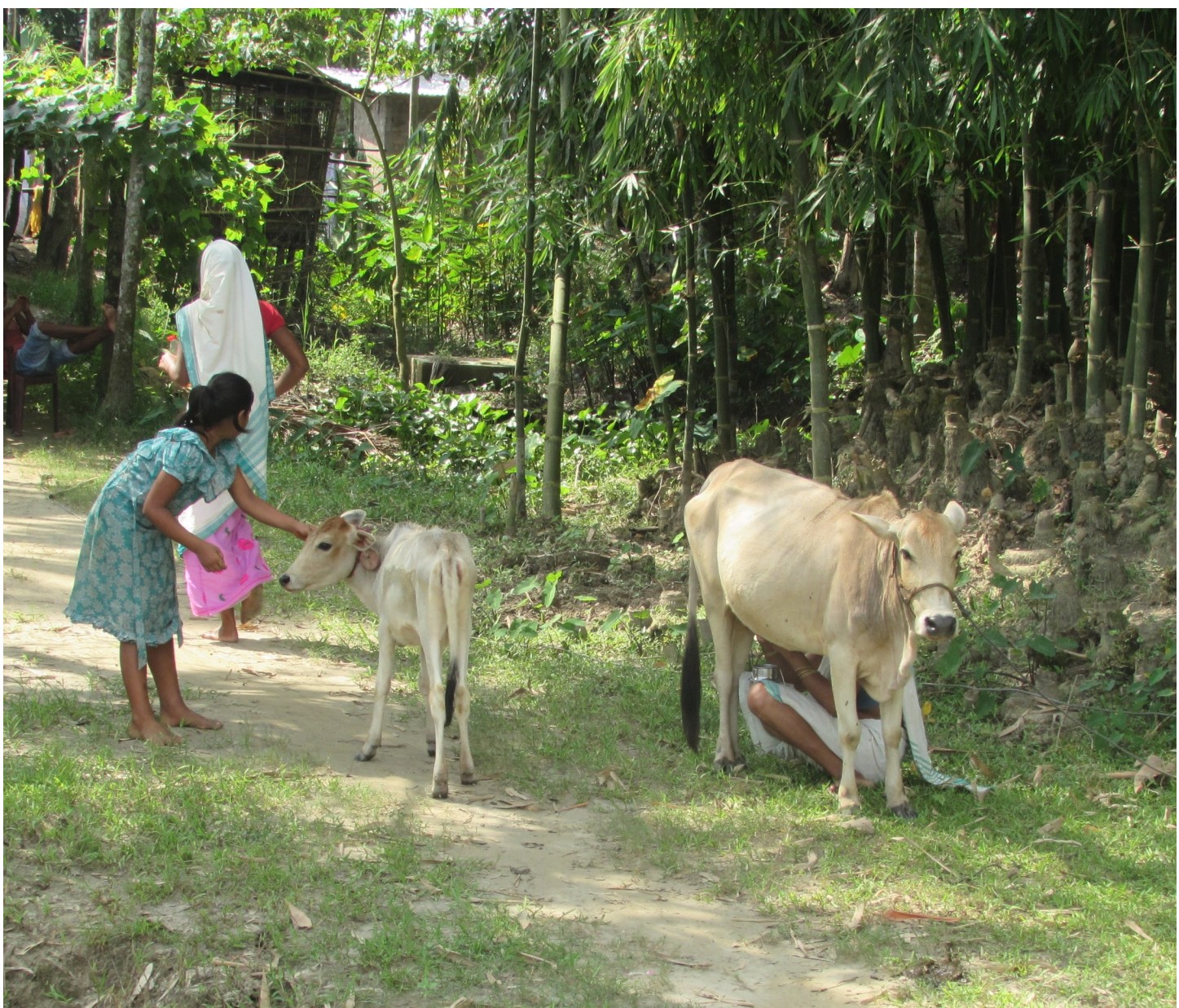

**Fig 4. Cows and calves are separated for daily milking.** The animals are brought out from the accommodation in the morning, where they have been separated overnight by a low barrier to prevent the calf from drinking. The calf initially drinks, and then remains close to the cow during milking by to stimulate let down. After milking the animals are taken to graze.

"*If dystocia occurs the farmer themselves corrects it, most people know about this. All the people in this village know about those things because the veterinary doctor is not always available. We last delivered a calf here at this house in 2008. My wife did it, she knows a lot more about these things than me.*"

(Participant 11, male, 58 years old)

"*If dystocia occurs call (veterinary) Doctor, there are no experienced men in this area.*"

(Participant 14, female, 34 years old)

"*Call the doctor if dystocia occurs.*"

(Participant 12, male, 35 years old)

"*If dystocia occurs- fetch . . . . . .'s wife. Usually, it is women who help animals to be born, they understand this work.*"

(Participant 13, male, 52 years old)

Participant 15, is a traditional and spiritual healer, as well as being a farmer. She also assists with the delivery of human babies.

"*When a cow starts labouring first push her belly down and in. First, rub the cow's belly, rub it when she heaves. Now we have to look (feel inside the birth canal) for the head. When you have it, pull together on the legs and head. You must pull in the same instant when she pushes. . . If the (calf's) leg is back it is very difficult to pull it out. If we cannot get it (the leg) to pull, then we call the doctor. If the calf is backwards then we always call the doctor.*"

(Participant 15, female, 45 years old.)

**Retained Foetal Membranes (RFM).**   In tropical conditions RFM can lead to both fly-strike and infection, as flies are rapidly attracted to all discharges and wounds. This can have a major effect on the cow's ability to nurse her calf and her productivity during lactation.

Participants discussed the importance of giving the cow warm water and/ or cooked food after giving birth, and there was an opinion that this helped to aid the passage of the placenta. A variety of foodstuffs were recommended by different households as beneficial in this regard.

Participant 15 described her technique for the treatment of RFM:

"*Sometimes the cow's placenta doesn't fall on its own and I take a bamboo stick and roll it up. I do this as quickly as possible- in between 2 days. I turn (the stick) and rock it and turn again. If not out (the placenta) in one day I call the doctor as it might cause severe damage inside the cow if not pulled out. If flies come, we apply haldi (*turmeric*) paste (*a natural antiseptic*). No herb or plant helps (the placenta to fall) but we give the cow warm water to drink. The calf drinking milk can help the placenta to fall out.*"

## Discussion

The attitudes of the respondents to their livestock can be characterised as their cows being family members who make an economic contribution to the family. Farmers respect and protect their animals, but also want the economic benefits of productivity. Animal keepers aspire to care for their animals to maximise health, welfare, and productivity. Shortcomings to this ideal stem from economic and educational causes, for example colostrum provision is suboptimal in part because of variable understanding of its importance; while milk provision for calves is frequently insufficient because of the family's need for milk for consumption and sale.

The study included in-depth interviews with 18 households, a sample which should not be assumed to be completely representative of all smallholder farmers in the locality, however the sample included participants of differing gender, age, wealth, social status, and education level, so a range of household circumstances and personal viewpoints were represented, and strong common themes found. Direct observation of farming practices, village life, and clinical examination supported the evidence of discussions with interview participants. It is reasonable to assume that some or all of these themes are important features in other villages in the region.

Respondents were very attentive to the care of their newborn calves. The majority of participants described allowing their calves to consume between 0.5 and 1 l of colostrum, usually starting between 1 and 2 hours after birth. This is better than the situation described in other studies on the Indian Subcontinent [17, 18, 19, 21]. However, the provision of colostrum for newborn calves in the study village is still below the target of 10–15% of bodyweight, and this has potentially far-reaching consequences for survivability, health, growth and long term productivity.

Newborn calves in the region typically weigh 15–20 kg, therefore ideal colostrum intake could be considered to be 2.25 to 3 l (15% of body weight) and adequate intake 1.5–2 l (10% of body weight), when calculated using figures quoted in literature. It is reasonable to assume that for reasons of genetics, diet and water intake, there is variation from the quoted figures for colostrum concentration for these native cattle compared to the intensively managed Holstein cows featured in the majority of scientific studies [30]. It is probable that native cattle produce more concentrated colostrum than high yielding Holsteins [31]. The diet of native cattle of heavily grazed pasture, rough foraging, and agricultural and vegetable waste, is of variable quality which might affect the quantity of IgG in colostrum [32, 33], particularly if the overall protein is low. The daily yields of colostrum and milk could also be decreased, particularly is energy intake is suboptimal, or daily water intake is low. Despite these unknowns, the provision of colostrum to calves, as described by respondents, was clearly insufficient in 10 out of 18 households, and questionable in a further seven households. If these participants are representative, colostrum provision for calves is insufficient in most households in the village.

The reason behind low colostrum intake of calves is primarily the local practice of taking colostrum and using it to make sweets. Unlike previous studies [17, 18, 19] only a few farmers related early colostrum feeding to neonatal diarrhoea or retained foetal membranes, though the statement *"yellow colour (of colostrum) (should be) faded in milk before calf drinks"* resonates with the findings of some authors investigating human neonatal health in other areas of India [20]. Investigations in the Kanha region of Madha Pradesh indicate that taking colostrum to make sweets is a normal practice in that region [24], and the practice known to occur throughout the Indian subcontinent [25]. There was no evidence in the study area of the sale of bovine colostrum as a human health food.

Sharing of food and produce is part of the social fabric of Assamese village life, building and maintaining bonds of community, which can be drawn upon in both happy times and challenging ones. Colostrum sweets do not seem to carry a specific social, religious, or health significance in this locality, but they are part of a wider tapestry of community and neighbourliness. It should be emphasised however that this statement cannot necessarily be extrapolated to other areas of India, where colostrum sweets could carry different cultural significance.

Navel care was not routinely undertaken in the region. Four respondents discussed navel care as important to prevent flystrike, but umbilical infection was not mentioned, unlike more intensive farming systems, where the open umbilicus is considered an important portal for the ingress of infection.

Current practices in smallholder milk production are inefficient both in terms of calf growth and milk yield, as a single cow has to provide both for the needs of her calf and the human household. There is no simple solution to this problem, and current milking practices that are followed after colostrum production has changed to milk are wholly reasonable in the local circumstances, to allow the cow to fulfil both of her roles. A nuanced, holistic and realistic approach needs to be taken to improving productivity, and the key to achieving this is farmer education, allowing animal keepers to keep ownership of the process of improvement.

Maximising the nutrition and water intake of the cow will help to maximise yields, and providing additional feeding to growing calves will improve growth and future productivity.

While this is difficult to achieve in a resource-limited environment, properly informed animal keepers are best placed to utilise available resources sustainably. Genetic potential limits the milk yields, growth rates, and fertility of animals. Improving cattle genetics can help to address this, however larger higher yielding animals also require more food and water to reach this potential, and must be sufficiently hardy to resist local environmental conditions and health challenges, thus programmes of genetic improvement must be carefully considered to be locally appropriate, and fully engage local animal keepers, who must be properly informed. Infectious disease reduces efficiency, productivity and survival, education of farmers of the benefits of a planned approach to reducing the incidence of disease through hygiene, vaccination, sourcing of animals, and managed grazing can yield further benefits through community–led approaches.

This study demonstrates a clear need to provide education to cattle keepers as to the importance of providing adequate colostrum as soon as possible after birth. Farmer education can take a variety of formats, however numerous factors must be taken into account to ensure that such a programme is both effective and achievable. Specific consideration should be given to the availability of farmers to attend educational meetings, and the location and timing of those meetings. The level of literacy of the intended recipient should be taken into account, as should their ability to interpret images, along with any local conventions in symbolism or image interpretation. The local situation should be properly investigated to ensure that educational content and materials are relevant and appropriate. Face to face interaction with local staff is invaluable for engaging animal keepers [24], and a train the trainer approach using local staff is essential for efficient delivery of education. Properly piloted, locally appropriate printed materials can form a valuable reference source following these engagements, allowing participants to review and refer to information at a later date.

There are strong cultural and socio-economic factors involved in colostrum and milk provision, and the position of the cow and calf as simultaneously part of the family unit, and contributors to the nutrition and income of the family, leads to a dichotomy of need. This intervention must be sensitive, respectful, and locally appropriate. Women and men are both involved in the care of livestock in most households, though women frequently take the greater role, particular with regards to calves and milking. Any education programme should be targeted to contact both equally. The high level of engagement of the animal keepers participating in this study indicates great opportunities for positive change.

Animal keeper education provided at a local level has great potential improve calf survival, growth and lifelong productivity, resulting in a positive impact on rural prosperity and efficient resource use. In particular, improving knowledge of the importance of colostrum provision to neonatal calves could yield sustainable benefits for minimal costs in this region.

## Supporting information

**S1 Data. Raw anonyomised interview data.**
(DOCX)

## Acknowledgments

The authors wish to thank the people of the villages of the Golaghat and Nagaon districts of Assam, without whose help and co-operation this project would not have been possible. We would also like to thank our colleagues at The Corbett Foundation and The University of Edinburgh for their help and support with this project.

## Author Contributions

**Conceptualization:** Andy Hopker, Naveen Pandey, Sophie Hopker, Neil Sargison, Rebecca Marsland.

**Data curation:** Andy Hopker.

**Formal analysis:** Andy Hopker, Naveen Pandey, Sophie Hopker, Rebecca Marsland.

**Funding acquisition:** Neil Sargison.

**Investigation:** Andy Hopker, Jadumoni Goswami, Rupam Saikia, Dibyajyoti Saikia.

**Methodology:** Andy Hopker, Naveen Pandey, Jadumoni Goswami, Sophie Hopker, Rebecca Marsland.

**Project administration:** Andy Hopker, Naveen Pandey, Jadumoni Goswami, Dibyajyoti Saikia, Neil Sargison.

**Resources:** Naveen Pandey, Rupam Saikia.

**Supervision:** Neil Sargison, Rebecca Marsland.

**Writing – original draft:** Andy Hopker.

**Writing – review & editing:** Andy Hopker, Naveen Pandey, Jadumoni Goswami, Sophie Hopker, Rupam Saikia, Amy Jennings, Dibyajyoti Saikia, Neil Sargison, Rebecca Marsland.

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
