## [Decision Letter · Decision Letter 0]

20 Nov 2019

PONE-D-19-26992

Colostrum provision and care of calves among smallholder farmers in the Kaziranga region of Assam, India.

PLOS ONE

Dear Mr Hopker,

Thank you for submitting your manuscript to PLOS ONE. After careful consideration, we feel that it has merit but does not fully meet PLOS ONE’s publication criteria as it currently stands. Therefore, we invite you to submit a revised version of the manuscript that addresses the points raised during the review process.

The study is interesting work and well presented, I feel that the manuscript is dealing with a good topic but lacks in the quality of preparation. I agree with reviewers, and please review the referee comments and make your peer revision.

We would appreciate receiving your revised manuscript by Jan 04 2020 11:59PM. To enhance the reproducibility of your results, we recommend that if applicable you deposit your laboratory protocols in protocols.io, where a protocol can be assigned its own identifier (DOI) such that it can be cited independently in the future. For instructions see: http://journals.plos.org/plosone/s/submission-guidelines#loc-laboratory-protocols

We look forward to receiving your revised manuscript.

Kind regards,

Arda Yildirim, Ph.D.

Academic Editor

PLOS ONE

Journal Requirements:

2. Your ethics statement must appear in the Methods section of your manuscript. If your ethics statement is written in any section besides the Methods, please move it to the Methods section and delete it from any other section. Please also ensure that your ethics statement is included in your manuscript, as the ethics section of your online submission will not be published alongside your manuscript.

3. We note that Figures [1 and 4] includes an image of a participant in the study. 

Additional Editor Comments (if provided):

For your guidance, you could check the reviewers' comments. Thank you for giving us the opportunity to consider your work.

Reviewers' comments:

Reviewer's Responses to Questions

**Comments to the Author**

1. Is the manuscript technically sound, and do the data support the conclusions?

Reviewer #1: Yes

Reviewer #2: Partly

2. Has the statistical analysis been performed appropriately and rigorously? 

Reviewer #1: Yes

Reviewer #2: N/A

3. Have the authors made all data underlying the findings in their manuscript fully available?

Reviewer #1: Yes

Reviewer #2: Yes

4. Is the manuscript presented in an intelligible fashion and written in standard English?

Reviewer #1: Yes

Reviewer #2: Yes

5. Review Comments to the Author

Reviewer #1: This is an interesting study. As someone who does work in India, I commend you for the time it would take to conduct this type of evaluation of calving and calf management practices. And as someone that works on calves, this is also interesting.

There are a few spelling and grammar mistakes throughout, but overall it is well written.

Please exchange may for might. The use of may implies asking permission, whereas something might happen.

Line 60: This is true only as an example. IgG levels can vary greatly due to the nutrition, environment and immunological status of the dam, so it is imperative in any system to have some way to measure immunoglobulins on a cow by cow basis. Please make a statement about that and there are many references in the literature.

264 start sentence with something other than a number.

Reference out of place

It would be helpful if you made a suggestion about what type of intervention and educational materials should be provided. Some of the farmers/people are illiterate from my experience which means educating them about science requires a different format of information exchange. What would you recommend?

Reviewer #2: This manuscript is interesting and useful to develop and plan regional dairy farm/industry. However, the author should present their results in a rather more objective manner. for example inadequate, notable, commonly practiced etc are not scientific wording. In addition, the author need to clarify types of cows.

6. PLOS authors have the option to publish the peer review history of their article (what does this mean?). If published, this will include your full peer review and any attached files.

Reviewer #1: No

Reviewer #2: Yes: Inchul Choi

---

## [Author Response · Author response to Decision Letter 0]

17 Dec 2019

Dear Arda Yildirim (Academic editor) and Reviewers

Thank you for the time taken to review our manuscript and for your positive and constructive comments.

We have attempted to address all of the points that you have raised, please find these detailed in an item by item fashion below 

This has been altered in accordance with the templates provided. Thank you.

2. Your ethics statement must appear in the Methods section of your manuscript. If your ethics statement is written in any section besides the Methods, please move it to the Methods section and delete it from any other section. Please also ensure that your ethics statement is included in your manuscript, as the ethics section of your online submission will not be published alongside your manuscript.

The ethics statement now appears beneath a separate subheading in the methods section.

3. We note that Figures [1 and 4] includes an image of a participant in the study. 

The narrative of the manuscript is of the co- existence and co- dependency of people and domestic animals, and we believe that the manuscript is greatly enriched for the addition of relevant images to illustrate this. These two images have been specifically chosen as not only do they illustrate key points, the human participants are impossible to identify as no details of faces are visible whatsoever. The individuals concerned did supply informed verbal consent, which was recorded, however, for the reasons explained in the manuscript written consent was not sought. We would request that as one cannot identify the individuals shown in the images you would consider accepting these images for publication. If it remains impossible to use these images, we will be able to alter the manuscript to include others, however we have specifically selected these images for their contribution to the richness of the manuscript, while preserving the anonymity of the participants. Thank you. 

Reviewer #1: This is an interesting study. As someone who does work in India, I commend you for the time it would take to conduct this type of evaluation of calving and calf management practices. And as someone that works on calves, this is also interesting.

There are a few spelling and grammar mistakes throughout, but overall it is well written.

Thank you for your kind and constructive comments. We have revised the manuscript to remove all errors that we have found and to improve the clarity of written English.

Please exchange may for might. The use of may implies asking permission, whereas something might happen.

We have amended the use of the word “may” throughout the manuscript to use terms such as “might”, “can” or “sometimes”. Thank you for bringing the need for these changes to our attention as this improves the clarity of the manuscript.

Line 60: This is true only as an example. IgG levels can vary greatly due to the nutrition, environment and immunological status of the dam, so it is imperative in any system to have some way to measure immunoglobulins on a cow by cow basis. Please make a statement about that and there are many references in the literature.

We have added a statement regarding factors resulting in variation in colostrum quality and a second statement on the value and challenges of assessing individual animals with supporting references. Thank you for raising this important point.

264 start sentence with something other than a number.

Done. Thank you.

Reference out of place

Attended to. Thank you.

It would be helpful if you made a suggestion about what type of intervention and educational materials should be provided. Some of the farmers/people are illiterate from my experience which means educating them about science requires a different format of information exchange. What would you recommend?

We have briefly expanded upon this point at the end of the discussion, with particular reference to a previous project carried out in Madhya Pradesh. Currently we are trialling an intervention in the locality, which will be analysed and reported next year.

Reviewer #2: This manuscript is interesting and useful to develop and plan regional dairy farm/industry. However, the author should present their results in a rather more objective manner. for example inadequate, notable, commonly practiced etc are not scientific wording. In addition, the author need to clarify types of cows.

Thank you for your positive and constructive comments.

The manuscript has been revised to replace these terms, and others, with more precise language. Thank you for bringing this to our attention as these changes have improved the manuscript.

Local cows are described in the manuscript line 215 – 217. The types of cows owned by the participants are now explicitly stated in the relevant section of the results, line 251- 253.

We hope that the revised manuscript now meets with your approval for publication. Should further alterations be required, please do not hesitate to contact us further.

Yours sincerely

Andy Hopker

---

## [Decision Letter · Decision Letter 1]

24 Jan 2020

Colostrum provision and care of calves among smallholder farmers in the Kaziranga region of Assam, India.

PONE-D-19-26992R1

Dear Dr. Hopker,

We are pleased to inform you that your manuscript has been judged scientifically suitable for publication and will be formally accepted for publication once it complies with all outstanding technical requirements.

With kind regards,

Arda Yildirim, Ph.D.

Academic Editor

PLOS ONE

https://www.researchgate.net/profile/Arda_Yildirim2

Additional Editor Comments (optional):

Thank you for responding to all comments and for revising the manuscript. Best wishes,

Reviewers' comments:

Reviewer's Responses to Questions

**Comments to the Author**

1. If the authors have adequately addressed your comments raised in a previous round of review and you feel that this manuscript is now acceptable for publication, you may indicate that here to bypass the “Comments to the Author” section, enter your conflict of interest statement in the “Confidential to Editor” section, and submit your "Accept" recommendation.

Reviewer #2: All comments have been addressed

2. Is the manuscript technically sound, and do the data support the conclusions?

Reviewer #2: Yes

3. Has the statistical analysis been performed appropriately and rigorously? 

Reviewer #2: Yes

4. Have the authors made all data underlying the findings in their manuscript fully available?

Reviewer #2: Yes

5. Is the manuscript presented in an intelligible fashion and written in standard English?

Reviewer #2: Yes

6. Review Comments to the Author

Reviewer #2: this manuscript reported that suitable provision of nutrition and care support local small farming, this kind of report will be useful to improve regional health and welfare to human and animal, This article was revised well and the authors addressed raised issues by reviewers.

7. PLOS authors have the option to publish the peer review history of their article (what does this mean?). If published, this will include your full peer review and any attached files.

Reviewer #2: Yes: Inchul Choi

---

## [Editor Report · Acceptance letter]

2 Mar 2020

PONE-D-19-26992R1 

Colostrum provision and care of calves among smallholder farmers in the Kaziranga region of Assam, India. 

Dear Dr. Hopker:

I am pleased to inform you that your manuscript has been deemed suitable for publication in PLOS ONE. Congratulations! Your manuscript is now with our production department. 

With kind regards,

on behalf of

Dr. Arda Yildirim 

Academic Editor

PLOS ONE